# Enrichment of Tumor-Infiltrating B Cells in Group 4 Medulloblastoma in Children

**DOI:** 10.3390/ijms23095287

**Published:** 2022-05-09

**Authors:** Kuo-Sheng Wu, Ting-Yan Jian, Shian-Ying Sung, Chia-Ling Hsieh, Man-Hsu Huang, Chia-Lang Fang, Tai-Tong Wong, Yu-Ling Lin

**Affiliations:** 1Graduate Institute of Clinical Medicine, School of Medicine, College of Medicine, Taipei Medical University, Taipei 110, Taiwan; abel1063@gmail.com (K.-S.W.); ssung@tmu.edu.tw (S.-Y.S.); 2Agricultural Biotechnology Research Center, Academia Sinica, Taipei 115, Taiwan; frogcty@gate.sinica.edu.tw; 3International Ph.D. Program for Translational Science, College of Medical Science and Technology, Taipei Medical University, Taipei 110, Taiwan; 4The Ph.D. Program for Translational Medicine, College of Medical Science and Technology, Taipei Medical University, Taipei 110, Taiwan; chsieh0404@gmail.com; 5TMU Research Center of Cancer Translational Medicine, Taipei Medical University, Taipei 110, Taiwan; 6Office of Human Research, Taipei Medical University, Taipei 110, Taiwan; 7TMU-Research Center of Urology and Kidney, Taipei Medical University, Taipei 110, Taiwan; 8Laboratory of Translational Medicine, Development Center for Biotechnology, Taipei 115, Taiwan; 9Department of Pathology, Shuang-Ho Hospital, Taipei Medical University, New Taipei City 235, Taiwan; 19225@s.tmu.edu.tw; 10Department of Pathology, School of Medicine, College of Medicine, Taipei Medical University, Taipei 110, Taiwan; ccllfang@tmu.edu.tw; 11Department of Pathology, Taipei Medical University Hospital, Taipei Medical University, Taipei 110, Taiwan; 12Pediatric Brain Tumor Program, Taipei Cancer Center, Taipei Medical University, Taipei 110, Taiwan; 13Neuroscience Research Center, Taipei Medical University Hospital, Taipei 110, Taiwan; 14Division of Pediatric Neurosurgery, Department of Neurosurgery, Taipei Neuroscience Institute, Taipei Medical University Hospital, Taipei Medical University, Taipei 110, Taiwan

**Keywords:** medulloblastoma, G4 subgroup, tumor-infiltrating B cells, mast cells

## Abstract

Medulloblastoma (MB) is the most common malignant brain tumor in children. It is classified into core molecular subgroups (wingless activated (WNT), sonic hedgehog activated (SHH), Group 3 (G3), and Group 4 (G4)). In this study, we analyzed the tumor-infiltrating immune cells and cytokine profiles of 70 MB patients in Taiwan using transcriptome data. In parallel, immune cell composition in tumors from the SickKids cohort dataset was also analyzed to confirm the findings. The clinical cohort data showed the WNT and G4 MB patients had lower recurrence rates and better 5-year relapse-free survival (RFP) compared with the SHH and G3 MB patients, among the four subgroups of MB. We found tumor-infiltrating B cells (TIL-Bs) enriched in the G4 subgroups in the Taiwanese MB patients and the SickKids cohort dataset. In the G4 subgroups, the patients with a high level of TIL-Bs had better 5-year overall survival. Mast cells presented in G4 MB tumors were positively correlated with TIL-Bs. Higher levels of CXCL13, IL-36γ, and CCL27 were found compared to other subgroups or normal brains. These three cytokines, B cells and mast cells contributed to the unique immune microenvironment in G4 MB tumors. Therefore, B-cell enrichment is a G4-subgroup-specific immune signature and the presence of B cells may be an indicator of a better prognosis in G4 MB patients.

## 1. Introduction

Medulloblastoma (MB) is a type of CNS embryonal tumor with high malignancy and is histologically defined into classic, desmoplastic/nodular (DNMB), MB with extensive nodularity (MBEN), and large cell/anaplastic types. MB with myogenic and/or melanotic differentiation is very rare [1,2]. Medulloblastoma constitutes 63% of the CNS embryonal tumors in children and adolescents (ages 0–19) in the United States [2]. As a result of a large cohort genomic study and clinical/biological heterogeneity, four principal subgroups of MB have been genetically defined as WNT activated (WNT), sonic hedgehog activated (SHH), Group 3 (G3), and Group 4 (G4) [3,4,5]. Non-WNT/non-SHH MBs comprise G3 and G4 tumors [1]. Within the four principal subgroups, further subgroups are categorized. There are four subgroups of SHH MB (α, β, γ, δ) [3] and eight subgroups of non-WNT/non-SHH MB (I-VIII) [4,5,6,7,8]. Clinically, these subgroups vary in demography, histological type, subgroup distribution, metastatic status, and survival. Non-WIN MB with metastasis (intracranial and/or intraspinal) at diagnosis is the most significant factor influencing survival. Other clinical factors include histological type (e.g., very high risk of recurrence in G3 tumors associated with LC/A), extent of tumor resection (subtotal versus with residue tumor >1.5 cm^2^), and age (ages 0–2 and ages 3–17) [2,6,7,8]. The most significant molecular prognostic factors include TP53 mutation and MYCN amplification in the SHH subgroup and Chr. 11 loss in G4 (as a good prognostic factor) [7,8,9,10].

G4 MB makes up the largest population of MB patients and commonly occurs in children and adolescents (ages 3–19). The overall survival of G4 MB in children is better than G3 MB [4,10]. In our reported cohort series of 56 MBs in children, the proportions of LC/A type and metastasis at diagnosis in G3 and G4 MBs were 7/15 (46.7%) and 6/13 (46.2%), and 6/15 (40%) and 4/13 (30.8%), respectively. Six out of the seven (85.7%) patients with LC/A in G3 MB had recurrence compared to two out of six (33.3%) patients in G4 MB [11]. In immune cell enrichment analysis of this series, we observed that the B cell population in G4 MB was more abundant than in the other three subpopulations. In this study, we evaluated the characteristics of TIL-B in the immune microenvironment of G4 MBs and the link with relatively good survival for these tumors.

The composition of the tumor microenvironment (TME) has been proven to be highly associated with prognosis and affects the incidence of refractory and recurrent disease in MB patients. Due to the small sample size and lack of molecular subgroup information, information on immune and stromal cells in the MB TME is still insufficient. Molecular subgrouping results indicate a correlation between M1 or M2 types of tumor-associated macrophage (TAM) recruitment and survival. CD163-positive M2 macrophages have been found in SHH MB, but high M1 macrophage recruitment has worse overall survival (OS) and progression-free survival (PFS) in SHH MB [11]. Transcriptional profiling showed G4 MB had more cytotoxic lymphocytes and neutrophils than other MB subgroups, according to tumor-infiltrating cell population scoring, whereas there was no significant difference between the four subgroups in total T and CD8-positive T cells [12]. Cytokines and chemokines modulate immune cell infiltration and cancer progression. Compared with the other subgroups, SHH tumors exhibited more inflammatory cytokines, such as IL-1β and CCL2 [12]. CCL2 is a neuroinflammatory cytokine that induces inflammation and a chemoattractant that recruits monocytes and dendritic cells to the injured tissues. The CCL2 in the cerebrospinal fluid (CSF) of metastasis patients is higher than in non-metastatic G3 and G4 MB [13]. CXCL1, IL-6, and IL-8 are highly expressed in the CSF of the MYC-amplified G3 MB patients [12]. The PD1/PD-L1 axis is considered to be a prospective target for immunomodulation in MBs. However, programed death-ligand 1 (PD-L1) had low levels of expression in all subgroups of MB [14]. Therefore, targeting the PD1/PD-L1 axis may not be suitable for MB treatment in virtue of its low expression.

Recently, tumor-infiltrating lymphocytes (TIL), especially CD4^+^ T cells, cytotoxic CD8^+^ T cells, and γδ-T cells have been associated with clinical benefit and favorable prognosis in the tumor microenvironment [15,16,17]. However, the role of tumor-infiltrating B cells (TIL-Bs) in inhibiting and enhancing the anti-tumor response is still controversial. B cells act as antigen-presenting cells to present tumor-associated antigen (TAA) to CD4^+^ T cells [18], and then help CD4^+^ and CD8^+^ T cells to secrete interferon-γ, resulting in immediate cytotoxicity to tumor cells [18,19]; meanwhile, B cells directly lyse tumor cells by secreting granzyme B and TRAIL [20]. Furthermore, antibody-producing B cells rejected tumor cells via the secretion of antibodies to induce antibody-dependent cell-mediated cytotoxicity (ADCC) and complement-dependent cytotoxicity (CDC). Plasma cells (CD138^+^ B cells) infiltrated with CD20^+^ B cells and CD4^+^ and CD8^+^ T cells in tumors improved survival of the patients by 65% at 10 years in ovarian cancer [21]. TIL-Bs behaved the immunoglobulin somatic hypermutation (SHM) and class-switch recombination (CSR) for the maturation of antibody response. In lung cancer patients, B cells expressed activation-induced cytidine deaminase (AID) to catalyze the SHM and CSR in tumor lesion tissue as well [22,23]. Thus, these studies demonstrated B cells are favorable for progression in the anti-tumor immune response.

On the other hand, some of the literature also shows that B cells are harmful to cancer prognosis. Adoptive transfer with B cells or anti-serum from K14-HPV16 mice into K-14-HPV16/Rag1−/− mice promoted epithelial carcinogenesis [24]. B cell infiltration and CXCL13 expression in the tumor area have a relationship with accelerated tumor growth and developed metastasis [25,26]. These B cells, which promote tumor growth, are classified to possibly be regulatory B (Breg) cells. Breg cells promote tumor growth due to their immunosuppressive function. IL-10 secreted by Breg cells attenuated the activation of T cells and NK cells to promote tumor progression [27,28]. Breg cells (CD19^+^ IL-10^+^ or CD19^+^ CD24^hi^CD38^hi^) increased the immune regulatory protein of PD-L1 and CTLA4 to promote the regulatory T (Treg) cells, myeloid-derived suppressor cells (MDSC), and tumor-associated macrophages (TAMs) [29]. Furthermore, Breg cells suppressed antitumor immunity by facilitating CD4^+^ T cells into Treg cells via TGF-β secretion [30,31]. Therefore, Breg cells promoted the survival of tumor cells by expressing high IL-10 and TGF-β to suspend anti-tumor immunity [32].

Although the G4 MB had high tumor metastasis at first diagnosis, the mortality rate of G4 MB was lower than the SHH group or G3 subgroups. Information about the composition of the tumor microenvironment and tumor-infiltrating immune cells is still unclear. In this study, we studied 70 patients with MB in Taiwan to analyze the tumor microenvironment and tumor-infiltrating immune cells through gene expression data. We also evaluated the SickKids cohort dataset to confirm our findings. We characterized the immune signature of MB patients and investigated the role of TIL-Bs in the G4 tumor. The unique immune signature of B cell enrichment in G4 tumors and the association with mast cells were also explored.

## 2. Results

### 2.1. Patient Cohort Clinical Data

The clinical data of the MB patients are summarized in Table 1. In this cohort series of 70 MBs in children, the median age at diagnosis was 6.4 years (ranging from 0.3 to 18.3 years). Fourteen children (20.0%) were ≤3 years and fifty-six (80.0%) children were >3 years. The male to female ratio was 1/1 (35/35). In total, 20 (28.6%) tumors presented with metastasis at diagnosis. Histological variants were classified as classic MB (*n* = 34, 48.6%), desmoplastic/nodular MB (*n* = 11, 15.7%), MB with extensive nodularity (*n* = 1, 1.4%), large cell/anaplastic (LCA) MB (*n* = 23, 32.9%), and MB with melanotic myogenic differentiation (*n* = 1, 1.4%). Tumor recurrence occurred in 27 (38.6%) cases. In a median follow-up period of 5.1 years (range, 0.3 to 25.5 years), the 5-year overall survival (OS) and relapse-free survival (RFS) of the whole cohort series were 74.1% and 63.0%, respectively. Among the four subgroups of MB, the WNT and G4 MB patients had lower recurrence rates and better 5-year relapse-free survival (RFS) compared with the SHH and G3 MB patients (Table 1).

### 2.2. Characterization of Tumor-Infiltrating Immune Cell Population and Subtypes of Tumor-Infiltrating B Cells in the MB Subgroups

To evaluate the abundance of the tumor-infiltrating immune cell population in the MB subgroups, we used xCell to perform enrichment analysis of 34 immune cell types from transcriptional profiling data. We studied the frequencies of T cells, B cells, macrophages, DCs, mast cells, and other myeloid cells in the tumor microenvironment from 70 cases of MB (Figure 1a). The WNT and SHH subgroups had higher macrophage and DC frequencies than the G3 and G4 subgroups (Figure 1a). There were increases in macrophage subsets (macrophages, M1 and M2 macrophages), DC subsets (DCs, aDCs, cDCs, and iDCs), and neutrophils in the WNT and SHH subgroups compared with the G3 and G4 subgroups (Figure 1b). SHH tumors had higher levels of expression of Th1 cells and CD8^+^ T cell subsets (CD8^+^ naïve T, CD8^+^ Tcm, and CD8^+^ Tem), but high levels of Tregs were also seen (Figure 1b). The G3 subgroup had the highest T cell frequency (Figure 1a). G3 tumors had more CD8^+^ T subsets (CD8^+^ Tcm and CD8^+^ Tm), Th1, and NKT cells compared to WNT and G4 tumors (Figure 1b). Among the MB subgroups, the G4 subgroup had the highest B cell frequency (Figure 1a). It is worth noting that all B cell subsets, including naïve B, memory B, class-switched memory B, pro-B and plasma cells in G4 MB, were more abundant than in the other three subgroups (Figure 1b). Figure 1c shows more details of the subtypes of the infiltrating B cells in MB tumors. We found that G4 tumors had the highest infiltration levels of total B cells, pro-B, naive B, class-switched memory B, and plasma cells. To estimate whether B cell enrichment is specific in G4 tumors, we also analyzed the MB public data from the SickKids cohort study. The clinical data of the MB patients from the SickKids cohort study are summarized in Appendix A. As shown in Figure 2a, an abundance of the B cell population was also seen in G4 tumors from SickKids cohort data. All subtypes of B cells in G4 tumors were higher than in the other three subgroups (Figure 2b). According to the 2021 WHO classification of tumors of the central nervous system, non-WNT/non-SHH medulloblastomas include G3 and G4, separated into eight subgroups. We also used the latest classification to study these 70 cases. The B cell populations in the VI and VIII subtypes of MB were similar to G4 tumors, which were more abundant than other subpopulations (Appendix A). In terms of this classification, we found class-switched memory B (*p* = 0.018) in MB VI/VIII tumors had higher infiltration than other subgroups (Appendix A). Thus, enrichment of B cell subsets should be a specific feature of the G4 MB tumor microenvironment.

### 2.3. Overall Survival Analysis of TIL-B in the G4 Subgroups

To investigate the role of TIL-B in G4 MB, we studied whether the higher TIL-B affects 5-year overall survival in G4 MB patients. Figure 3 shows that high levels of TIL-B, including naïve, memory, and plasma cells in G4 tumors, were associated with better 5-year overall survival compared to low levels of TIL-B in G4 tumors. The presence of B cells in the tumor may increase 5-year overall survival in G4 MB patients.

### 2.4. The Correlation of Immune Cells with B Cells in the G4 Subgroups

To characterize the TIL-B enrichment of G4 MB tumors, we examined the immune cells to understand the relationship with B cells. Although CD4^+^ T cells, CD8^+^ T cells, pDCs, and eosinophils were highly expressed in tumors, as shown in Figure 1b, there was no correlation with TIL-B (Figure 4). There was also no correlation between TIL-B and NK cells (R = 0.22, *p* = 0.38, Figure 4), and Th2 cells (R = 0.094, *p* = 0.71, Figure 4). Th1 cells were negatively correlated with B cells in G4 (R = −0.53, *p* = 0.023, Figure 4). Notably, mast cells were positively correlated with B cells in G4 (R = 0.78, *p* < 0.001, Figure 4). This result was similar to group VI and VIII MB in the 2021 MB classification (R = 0.89 and *p* < 0.05, Appendix A). Therefore, the enrichment of B and mast cells contributed to the G4 subgroup-specific immune feature.

### 2.5. Cytokine and Chemokine Profiling in the MB Subgroups

Cytokines and chemokines modulate immune cell infiltration and serve their function in the tumor microenvironment. There were no significant differences in the expression of Th2 cytokines, such as IL-4, IL-5, and IL-13, among the four subtypes of MB (Appendix A). In G4 MB tumors, the expression of IL-10, also produced by Th2 cells, was lower than that of the WNT and SHH MB tumors (Appendix A). The expression of IL-23A, IL-32, IL-36γ, and IL-37 in the G4 tumors was higher than that in normal brain tissue (Figure 5). In terms of chemokines, we found that the expression of CCL14, CCL27, and CXCL13 in G4 was higher than in the normal brain (Figure 6). CXCL13 is a well-known chemoattractant for B lymphocytes and has a high expression in G4, which is consistent with the B cell infiltration in G4 tumors. Thus, IL-36γ, CCL27, and CXCL13 expression in G4 was higher than in the normal brain.

### 2.6. TIL-B Subsets in the SHH and G4 MB Subgroups

The SHH and G4 MB sections were labeled for CD19, CD20, CD27, or CD138. Representative images of CD19^+^ B cells, CD20^+^ B cells, CD27^+^ memory B cells, and CD138^+^ plasma cells in the SHH and G4 MB sections are shown in Figure 7. The results showed that higher levels of B cells, memory B cells, and plasma cells were in the G4 MB tumoral and non-tumoral regions. This higher TIL-B of clinic pathological features was in the G4 MB sample cohort, in agreement with our transcriptome molecular results (Figure 1).

## 3. Discussion

Medulloblastoma is the most common malignant brain tumor in children. In this study, we investigated tumor-infiltrating immune cells and cytokine profiles in tumors from 70 MB cases in Taiwan. At the same time, we also evaluated the public SickKids cohort dataset to confirm our findings. We found that B cells were enriched in the G4 tumor, in both cases, from Taiwan and the SickKids cohort studies (Figure 1 and Figure 2). Patients with high levels of TIL-B had better 5-year overall survival compared with those with low levels of TIL-B (Figure 3). TIL-B had a positive correlation with survival and better prognosis in several cancer types [33,34,35,36,37,38,39,40,41,42,43]. TIL-B is thought to be positively associated with improved cancer outcomes due to the tertiary lymphoid structure (TLS). The heterogeneity of cellular components and intratumoral location in the TLS, its antitumor immunity, and clinical outcomes can all be affected. The presence of B cells, CD4^+^, CD8^+^ T cells, and DCs in the G4 tumor presumably characterize the TLS (Figure 1b). TLS co-infiltrating CD8^+^ T cells and CD20^+^ B cells enhanced survival in metastatic melanomas co-expressing CXCR5 and CXCL13, whereas B-cell-enriched melanomas were accompanied by increased levels of naive or memory T cells [35]. B cells can produce tumor-reactive antibodies to promote antibody-dependent cell-mediated cytotoxicity, antibody-dependent cellular phagocytosis, and complement-dependent cytotoxicity by natural killer cells, macrophages, and complement, respectively [16,17,18]. B cells also act as an antigen-presenting cell to activate CD4^+^ and CD8^+^ T cells [19,20,21]. The presence of B cells in the tumor is the most important prognostic factor, even the CD8^+^ T cells and cytotoxic components primed in the TME [37]. This may explain why G4 MB patients with high levels of TIL-B have better 5-year overall survival.

The association between B cells and mast cells is well known in IgE-mediated allergic reactions at mucosal sites, as well as in lymphoid and vascular tissues. The communication between B cells and mast cells is engaged by cytokines (mainly type 2 cytokines, IL-10, IL-6, and IL-33), membrane-bound receptors and ligands (e.g., CD40/CD40L), and granule products (histamine and proteases). These interactions can promote B cell proliferation, survival, and class-switch [44]. However, the relationship between B cells and mast cells in the tumor microenvironment is unclear. In G4 tumors, mast cell populations were positively correlated with B cells (Figure 4). IL-6, IL-10, and IL-33 were no different or lower than other MB subgroups (Appendix A). These results indicated that B and mast cells might interact directly through CD40/CD40L and/or CD70/CD27 [44]. Mast cells in tumors exhibit pro- and anti-tumorigenic activity on tumor initiation and development. In the role of anti-tumor immunity, mast cells secreted CCL3 and CXCL10 to recruit NK cells, CD4^+^ helper T cells, and CD8^+^ cytotoxic T cells [45,46]. In our results, we observed higher levels of CCL3 and CXCL10 in G4 than in the normal brain, but no significant difference among other MB subtypes (Appendix A). Other cytokines, such as CCL2 and CCL5, that modulate the pro-tumorigenic or anti-tumorigenic phenotypes of mast cells were also no different among the four subtypes (Appendix A). Therefore, the results of this study cannot identify the role of mast cells in G4 tumors as a pro- or anti-tumor phenotype, but they should contribute to the proliferation and accumulation of B cells in the tumor [47].

CXCL13, IL36γ, and CCL27 expression in the G4 tumor was higher than in the normal brain and other MB subgroups (Figure 4 and Figure 5). CXCL13, also previously named B-cell-attracting chemokine 1 (BCA-1), is a homeostatic chemokine that is constitutively expressed by stromal cells in B-cell-rich areas of secondary lymphoid tissues, such as the spleen, lymph nodes, and germinal centers in Peyer’s patches [48]. It has been reported that TIL-CD4^+^ and some TIL-CD8^+^ T cells are major CXCL13 producers in tumors. CXCL13 potentially promotes TLS formation and is correlated with B cell infiltration and germinal center maturation at the tumor site [49]. In addition to recruiting B cells, CXCL13 is a chemoattractant associated with DCs, naive T cells, and follicular helper T cells. The CXCL13-CXCR5 axis induces the migration of follicular helper T cells and B cells to the lymph node follicles through IL-21 stimulation [50]. CXCL13-producing follicular helper T cells promote B cell maturation in breast cancer tumors, which may be a key factor in activating de novo antitumor humoral responses [49]. IL-36γ is an extended interleukin of the interleukin 1 superfamily from keratinocytes or epithelial cells, which is secreted and induced the highly inflammatory form of programmed cell death by the Toll-like receptor ligands, such as polyinosinic-polycytidylic acid. [51]. Several studies showed that IL-36γ plays a critical role in anti-tumor immunity. IL-36γ promotes the antitumor immune responses by inducing type 1 lymphocytes, such as CD8^+^, NK, and γδ-T cells [52]. Moreover, IL-36γ converts macrophages into M1-like macrophages and IL-36γ has a positive correlation with the density of CD20^+^ B cells within TLS in tumors, indicating that local IL-36γ supports TLS formation within the TME [53]. Higher expression of IL36γ suggests better prognosis and longer survival in human hepatocellular carcinoma [54]. CCL27 (cutaneous T cell-attracting chemokine, CTACK), expressed by the keratinocytes, is a very important chemokine for T cell skin homing and maintains a minimum amount in normal skin [55,56]. CCL27 allows CCR10-expressed Ab-secreting B cells in circulating and mucosal epithelial tissues to elicit greater antigen-specific IgA [57].

IL-22 and IL-12B expression in G4 tumors was higher than in the SHH tumors (Appendix A). IL-22, a member of the IL-10 family, has multiple roles in host defense and epithelial homeostasis [58]. IL-22 induces tumor-elicited inflammation from the T lymphocytes (CD4^+^ T cells) production via the activation of STAT3 [59]. It has been reported that lack of IL-22 production affects B cell immunity. IL-22 expression within tertiary lymphoid organs contributes to the production of the lymphoid chemokines, CXCL13 and CXCL12, which, in turn, orchestrate B-cell clustering, lymphoid aggregation, and autoantibody production [60]. IL-12B is also named natural killer cell stimulatory factor 2 or cytotoxic lymphocyte maturation factor 2, which is a growth factor for activated T and NK cells. IL-12B is secreted from the antigen-presenting cells, such as the activated macrophages, B cells, or dendritic cells, and stimulates activated T-cell proliferation [61]. IL-12B has an anti-tumoral effect as the main driver of Th1-type immunity through initiating and amplifying IFN-γ production [61,62,63,64]. Th1 cells, including Th2 cells, have abilities to help B cells. Th1 cells migrate into B cell follicles to promote CD154-dependent B cell clonal expansion and Ab production [65]. Therefore, IL-22 and IL-12B should be instrumental for B cell activation in the G4 tumors.

The previous study showed that G4 tumors had the highest CD8^+^ T cells and neutrophils compared to other subgroup tumors [12]. However, the previous study shows G4 and G3 tumors had more B lineages compared to WNT and SHH tumors but did not reach a significant difference [12]. In our cohort study, the higher levels of CD8^+^ T cells and neutrophils were similarly found in the G4 subgroup compared to the SHH and G3 sub-groups (Figure 1b). Moreover, G4 tumors had the highest B cell subsets in our cohort study (Figure 1b,c). The differences in B cell results between our cohort and the previous study may be related to the number of patients. A small number of patients is the limitation of this study. We collected 70 MB patients in Taiwan from 1989 to 2019 and summarized B cells enrichment in 18 G4 patients. To investigate whether B cell enrichment is also presented in the G4 patients globally, we analyzed the SickKids cohort in addition and summarized our unique findings on B cells enrichment.

## 4. Materials and Methods

### 4.1. Patient Cohort

This cohort comprised 70 MB cases age <20 years at diagnosis who presented to Taipei Veterans General Hospital (Taipei VGH) and Taipei Medical University Hospital (TMUH) between 1989–2019. This cohort comprised 64 primary tumors, 5 first recurrence, and 1 metastasis. All subjects gave written informed consent in accordance with the Declaration of Helsinki. The samples were fully encoded and used under a protocol approved by the Institutional Review Board of Human Subjects Research Ethics Committee of the Taipei Medical University Hospital and Chang Gung Memorial Hospital, Taiwan (IRB approval number 201701441A3).

### 4.2. Review of Clinical Data

Clinical data and tumor images were retrospectively retrieved and reviewed from medical records and reports. Histological variants were classified according to the WHO 2007 classification [1]. The date at diagnosis was defined as the date of first tumor resection. The retrieved clinical data included age, sex, metastasis status, histological variant, follow-up, and death. We defined the status of metastasis at diagnosis as M0–1 and M2–3 according to Chang’s operative staging system [66].

### 4.3. Gene Expression Profiles and Subgroup Classification by RNA-Seq

RNA-Seq was performed in this cohort of 70 MB cases as described in a previous study [8]. Briefly, RNA-Seq was run in a Nextseq 500 sequencing instrument (Illumina) for paired-end reads. The RNA-Seq data of this cohort are available in Gene Expression Omnibus (GSE143940 and GSE158413). RNA-Seq raw data were aligned by Kallisto [67] and the gene expression table was extracted by the tximport [68] package in R environment. For clustering, unsupervised clustering analysis was performed based on the 10,000 most differentially expressed genes using the consensus clustering default parameters by Rtsne and validated by 22 subgroup-specific signature gene expression levels [69]. RNA-Seq results were sent to Taylor’s laboratory at the Hospital for Sick Children, Toronto, to help with counterpart clustering.

### 4.4. Immune Cell Enrichment Analysis

Immune cell deconvolution was estimated with RNA-Seq profiling by xCell [70]. The scores for 64 cell types in 5 major cell populations across lymphoid, myeloid, stem, stromal, and other cells were computed with the gene expression data set normalized to TPM level of 489 cell-population-specific markers. The scores which presented in arbitrary units of 34 cell types in lymphoid and myeloid were compared in MB subgroups and subtypes. The associations between categorized variables were determined by Kruskal–Wallis test. A *p* value of less than 0.05 was considered as statistically significant.

### 4.5. DNA Methylation Array Profiling

Illumina Infinium MethylationEPIC array was performed for 66 MB cases in this cohort series as described in a previous study [8]. Raw data files were read and preprocessed using minfi [71] and ChAMP [72] package in R environment. For clustering, unsupervised clustering analysis was performed based on the 10,000 most differential probes using consensus clustering and validated by 11 subgroup-specific signature probes [73].

### 4.6. Similarity Network Fusion Analysis for WNT and SHH Subtype Clustering

The similarity network fusion (SNF) method was performed in the cohort series with a combination of gene expression and DNA methylation data. Subtype clustering was performed based on the top 1% of the most differentially expressed common genes (*n* = 216) and probes (*n* = 3211) from a previous study [74]. The selected genes and probes list from a published study were acquired for subtype clustering analysis run by SNFtool package in the R environment [74]. The parameters of SNF from a previous study were modified as follows: WNT: K = 6, alpha = 0.6, T = 50, SHH: K = 15, alpha = 0.6, T = 50.

### 4.7. Random Forest Classifier for Non-WNT/Non-SHH Subtype Clustering

To predict subtypes of non-WNT/non-SHH MBs, a web-based random forest (RF) classifier of MB G3/4 subgroups (https://www.molecularneuropathology.org/mnp, accessed on 14 Aug 2021) which were previously reported was applied in this cohort [75]. Briefly, MethylationEPIC array raw signal IDAT-files were uploaded and normalized by a two-factor linear model on log^2^ transition to the classifier with adjustment for frozen derivatives and patient gender. The most differential 10,000 probes were implemented to calculate RF score between 0 and 1 with a threshold of ≥0.9 for non-WNT/non-SHH subtype prediction.

### 4.8. Immunohistochemistry (IHC) Analysis

All the surgically resected biopsy samples were fixed with 10% neutral buffered formalin, embedded in paraffin, and serially sectioned at 4 μm. The infiltrating B cell subsets in tumors and non-tumors were analyzed by immunohistochemistry staining. Tissue sections were stained with primary antibodies for CD19 (1:10, Bio SB, Goleta, CA, USA), CD20 (ready-to-use, Scy tek, Logan, UT, USA), CD27 (1:250, GeneTex, Hsinchu City, Taiwan), and CD138 (ready-to-use, Roche Ventana, Oro Valley, AZ, USA) and secondary antibodies, and visualized with diaminobenzydine (DAB, Roche Ventana, Oro Valley, AZ, USA).

### 4.9. Survival Analysis

Overall survival (OS) analysis was calculated by the date of first tumor surgery (diagnosis date), last follow-up, and death. Survival analysis based on the scores of various cell types was performed using the Kaplan–Meier method by using survminer package in R environment. The differences in survival were assessed using the log-rank test. The association between categorized variables was determined by Kruskal–Wallis test. A *p*-value of less than 0.05 was considered as statistically significant.

## 5. Conclusions

In this study, we characterized TIL-B and the unique immune microenvironment in G4 MB tumors. It was ascertained that the enrichment of B and mast cells contributes to the G4 subgroup-specific tumor signature. High expression levels of B cells in tumors increased 5-year overall survival in G4 MB patients. Therefore, the presence of B cells may be an indicator of a better prognosis in G4 MB patients.

## Figures and Tables

**Figure 1 ijms-23-05287-f001:**
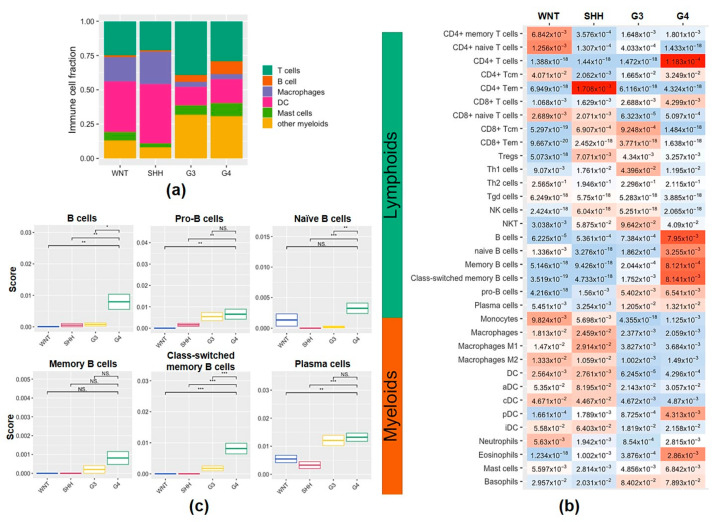
Analysis of immune cells and B cell subsets from the 70 MB patients. (**a**) The proportion of T cells, B cells, macrophages, DCs, mast cells and other myeloid cells in MB tumors. (**b**) The distribution of lymphoid and myeloid cell enrichment. Expression values are represented as colors and range from red (high), white (moderate) to light blue (low). (**c**) The expression of the infiltrating B cells subsets from in MB subgroups. *, *p* < 0.05; **, *p* < 0.01; ***, *p* < 0.001, compared to G4 subgroup tumors.

**Figure 2 ijms-23-05287-f002:**
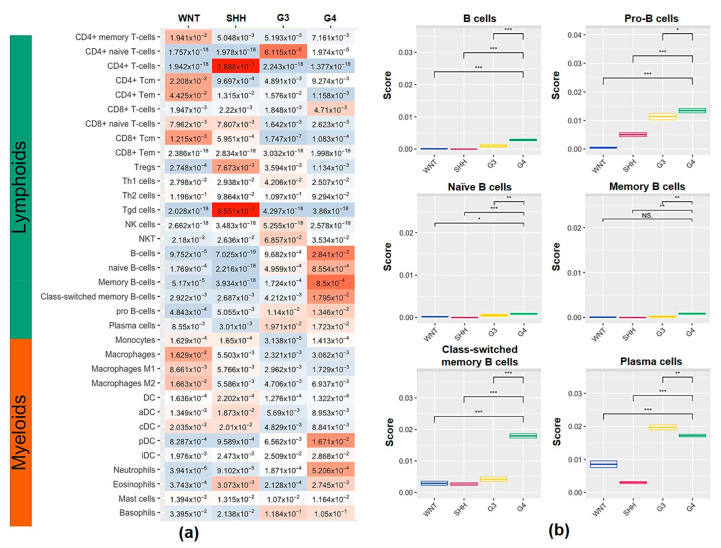
Analysis of immune cells and B cell subsets from the SickKids cohort study. (**a**) The distribution of lymphoid and myeloid enrichment from the SickKids cohort study (*n* = 763). Expression values are represented as colors and range from red (high), white (moderate) to light blue (low). (**b**) The expression of the infiltrating B cells subsets from in MB subgroups. *, *p* < 0.05; **, *p* < 0.01; ***, *p* < 0.001, compared to G4 subgroup tumors. NS: no significant difference.

**Figure 3 ijms-23-05287-f003:**
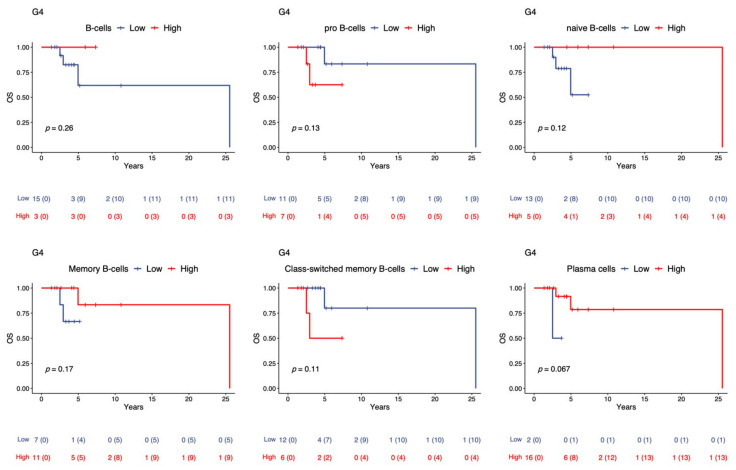
Kaplan–Meier overall survival curve of MB patients with high or low B cell expression. OS curves for patients according to the high and low expression levels of different B cell subsets, such as pro-B, naïve B, memory B class-switched memory B and plasma cells in G4 MB tumors.

**Figure 4 ijms-23-05287-f004:**
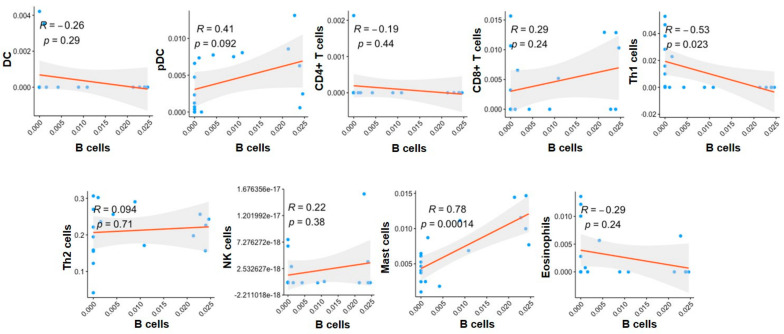
Correlation of B cells with other immune cell types in the G4 MB tumors. Correlation between the expression of B cells and the expression of DCs, pDCs, CD4^+^ T cells, CD8^+^ T cells, Th1 cells, Th2 cells, NK cells, mast cells and eosinophils in the G4 tumors. Correlation coefficients (R) were obtained by using Pearson’s correlation coefficient test. Blue dots indicate patients in the G4 subgroup.

**Figure 5 ijms-23-05287-f005:**
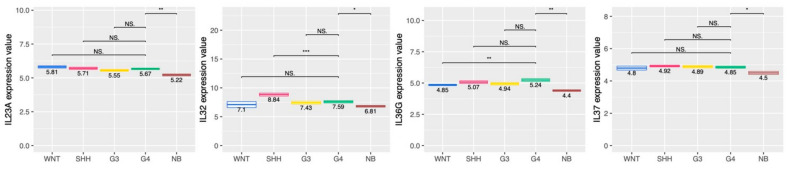
Expression of IL-23A, IL-32, IL-36γ, and IL-37 in different MB subgroup tumors and normal brain. The expressions of IL-23A, IL-32, IL-36γ, and IL-37 among 4 subgroups of MB tumors and normal brain tissue were analyzed from the transcriptome data. *, *p* < 0.05; **, *p* < 0.01; ***, *p* < 0.001, compared to G4 subgroup tumors. NS: no significant difference.

**Figure 6 ijms-23-05287-f006:**
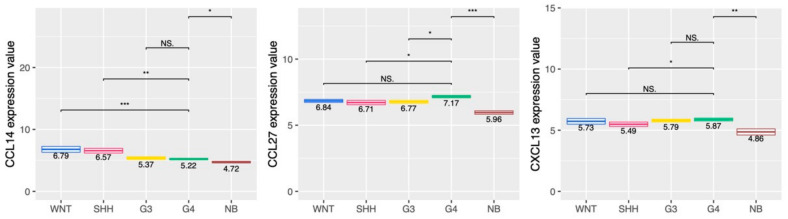
Expression of CCL14, CCL27, and CXCL13 in different MB subgroups, tumors and normal brain. The expressions of CCL14, CCL27, and CXCL13 among 4 subgroups of MB tumors and normal brain tissue were analyzed from the transcriptome data. *, *p* < 0.05; **, *p* < 0.01; ***, *p* < 0.001, compared to G4 subgroup tumors. NS: no significant difference.

**Figure 7 ijms-23-05287-f007:**
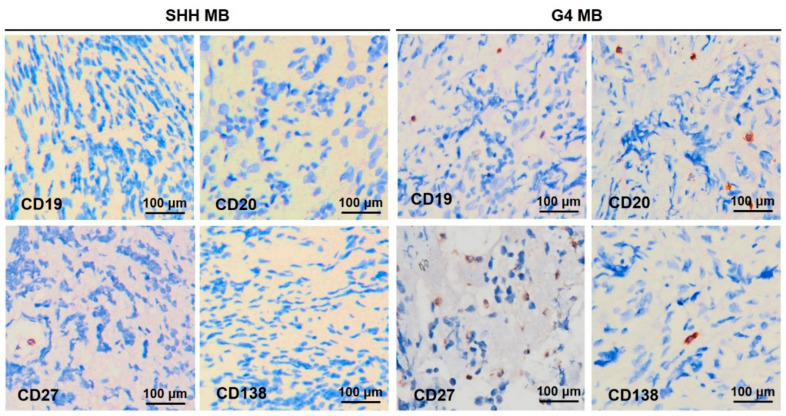
Representative photomicrographs of TIL-B subsets in patients with the SHH or G4 MB. IHC using primary antibodies against CD19, CD20, CD27, and CD138 to characterize TIL-B subsets in tumor tissue (original magnification, 400×).

**Table 1 ijms-23-05287-t001:** Demography, molecular subgroup, clinical data, and molecular–clinical correlation in our cohort of 70 childhood medulloblastomas (MBs) in Taiwan.

Molecular Subgroup Assignment				
*n* = 70	WNT *n* = 8 (11.4%)	SHH *n* = 24 (34.3%)	Group 3 *n* = 20 (28.6%)	Group 4 *n* = 18 (25.7%)
Frozen tissue for molecular profiling				
Primary tumor	8 (100%)	23 (95.8%)	17 (85.0%)	17 (94.4%)
Recurrent tumor	0	1 (4.2%)	3 (15.0%)	1 (5.6%)
Age (median, range) at diagnosis (years)				
6.4 (0.3–18.3)	7.9 (3.1–11.4)	4.1 (0.3–14.3)	5.1 (1.5–18.2)	10.1 (5.1–18.3)
≤3 y (*n* = 14, 20.0%)	0	8 (33.3%)	6 (30.0%)	0
>3 y (*n* = 56, 80.0%)	8 (100%)	16 (66.7%)	14 (70.0%)	18 (100%)
Sex				
Male, *n* = 35 (50.0%)	1 (12.5%)	12 (50.0%)	11 (55.0%)	11 (61.1%)
Female, *n* = 35 (50.0%)	7 (87.5%)	12 (50.0%)	9 (45.0%)	7 (38.9%)
Male/female ratio (1/1)	0.1/1	1/1	1.2/1	1.6/1
Metastasis stage at diagnosis (M0–1, M2–3), number of cases (percentage)				
M0–1, *n* = 50 (71.4%)	8 (100%)	20 (83.3%)	10 (50.0%)	12 (66.7%)
M2–3, *n* = 20 (28.6%)	0	4 (16.7%)	10 (50.0%)	6 (33.3%)
Pathology variant, number of cases (percentage) and median age (years)				
Classic, *n* = 34 (48.6%), 6.1	5 (62.5%),	12 (50.0%)	9 (45.0%)	8 (44.4%)
DNMB, *n* = 11 (15.7%), 3.5	1 (12.5%)	9 (37.5%)	0	1 (5.6%)
MBEN, *n* = 1 (1.4%), 1.0	0	1 (4.2%)	0	0
LCA, *n* = 23 (32.9%), 7.3	2 (25.0%)	2 (8.3%)	10 (50.0%)	9 (50.0%)
MMMB, *n* = 1 (1.4%), 4.3	0	0	1 (5.0%)	0
Recurrence, number of cases (percentage)				
Recurrence, *n* = 27 (38.6%)	1 (12.5%)	11 (45.8%)	11 (55.0%)	4 (22.2%)
Median follow-up time (range) (years)				
5.1 (0.3–25.5)	4.9 (2.8–11.7)	6.5 (0.7–17.7)	6.7 (0.3–12.5).	4.2 (1.4–25.5)
Survivals of molecular subgroup (percentage)				
5-year OS rate: 74.1%	100%	74.1%	62.4%	73.8%
5-year RFS rate: 63.0%	87.5%	62.5%	45.0%	68.5%

DNMB: Desmoplastic/nodular medulloblastoma, MBEN: Medulloblastoma with extensive nodularity, LCA: Large-cell/anaplastic, MMMB: Medulloblastoma with melanotic myogenic differentiation, M0–1: no metastasis of tumor cells in CSF, M2: intracranial subarachnoid space or intracranial compartment metastasis, M3: intraspinal subarachnoid space metastasis, OS: Overall survival, RFS: Relapse-free survival.

## Data Availability

RNA-seq data are available in Gene Expression Omnibus (GSE143940 and GSE158413).

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
