# Peer review of "Enrichment of Tumor-Infiltrating B Cells in Group 4 Medulloblastoma in Children"

_ijms, 2022, doi:10.3390/ijms23095287_

Round 1

Reviewer 1 Report

In this study, Wu et al. analyzed the type of immune cell infiltrates and the chemokine/cytokine profiles in a cohort of 70 medullobastoma patients using transcriptome data.

They found that tumor-infiltrating B cells were enriched in G4 subgroups of medulloblastoma and that the B cell levels were associated with better 5-year overall survival.

Although this is an interesting study, there are several issues which should be addressed:

Major points:

Whereas the cell type enrichment analysis from transcriptome profiling seems to be very robust and is based on their own data set and an even larger dataset from the SickKids cohort, survival analysis only includes a rather small cohort of 18 tumors in the G4 subgroup.

To strengthen the data and to see if there are significant differences between G4 MBs with high and low B-cell infiltrates, it would be important, if possible, to include survival data from the SickKids cohort.

Are there any clues why previous analysis of G4 MBs did not find increased B-cell infiltrates, but higher levels of cytotoxic CD8 T-cells and neutrophils (Diao et al. Oncology letters 2020)?

The authors did not find significant correlations between tumor-infiltrating B cells and CD4+ T-cells, CD8+ T-cells, NK-cells and DCs. Did the authors also include analysis of TH1(IL-12) or TH2 (IL-10) cells, pDCs and eosinophils? Again, is it possible to include data from the SickKid cohort?

The authors claim that tumor-infiltrating B cells in G4 MBs are associated with tertiary lymphoid structures. Do the authors have evidence from literature that such TLSs exist in G4 MB? To further strengthen their data, immunohistochemistry of G4 MBs with high levels of B-cells (non-SHH) in comparison to SHH MBs should be done to demonstrate the existence of such structures (CD19, CD20, CD27, CD138, CXCL13)

Regarding the cytokine/chemokine profiles in MB subgroups it would be more informative to compare G4 and SHH subgroups than normal brain. As shown in Figs. 5+6 CXCL13 and CCL27 expression was higher in G4 MBs than SHH MBs. Looking at Supl.FigS4 similar results were obtained for IL-22 and IL12B. Both cytokines might be even more important than IL-36. Can the authors speculate on this, especially the role IL-12B in this setting? Any association between Th1 cells and B-cells?

Minor points

Limitations of the study should be mentioned in the discussion.

The introduction is too long and should be more concise.

Reviewer 2 Report

The study “Enrichment of tumor-infiltrating B cells in group 4 medullo- blastoma in children” by Wu et al. examines the immune cell population among different subgroups of medulloblastoma tumors. They find that B cell infiltration is most prominent in Group 4 tumors among their 70 tumors (64 primary, 5 first recurrence , 1 metastasis). They further find that B cell infiltration is a good prognostic factor. They also compare their findings to a dataset from Michael Taylors group from Sick Kids in Toronto, Canada and can validate their findings.

Overall, the paper has an interesting finding and further highlights the differences among the different medulloblastoma (MB) subgroups. It will be interesting to validate those findings in high dimensional flow cytometry/ Immunohistopathology analysis of primary samples in further studies. This will strengthen the finding.

I do have some minor comments:

Introduction:

-may be better to give survival of all different subgroups, rather than stating better survival compared to Group 3 (Line 61)

- What does the statement mean “ THE TME has been proven to be associated with poor prognosis”? Is this true for any immune cell infiltration? Doesn’t it depend on the composition?(Line 70)

-TME has been proven rather than proved (Line 70)

-Please include definition of a Breg cell

Results:

Please clarify/expand: What are the differences in frequencies among subtypes?How many immune cell subsets overall?(Line 157)

-How many patients are in the Sick Kids Kohort? Can you please include a table with patient characteristics to see if they match your cohort?

-Figure Graphs seem blurry, please make sure the quality is good

-What does this mean? Hos is high vs. low B cell infiltration defined? Please include explanation how cut off was chosen? Also, is this true also for the other subgroups? Can you find high B cell infiltration in the other subgroups? How does this correspond to survival? Or is this only true for Group 4 tumors?

“Figure 3 shows that high levels of TIL-B,  including naïve, memory, and plasma cells in G4 tumors were associated with better 5- year overall survival compared to low levels of TIL-B in G4 tumors.” (Line 197-198)

Round 2

Reviewer 1 Report

No further comments. All my issues raised have been answered by the authors.